# Filling History, Consolidating the Origins. The First Female Architects of the Barcelona School of Architecture (1964–1975)

**Zaida Muxí** [1,*] and **Daniela Arias Laurino** [2,*]

1   ETSAB Barcelona School of Architecture, Universitat Politècnica de Catalunya, 08034 Barcelona, Spain
2   Department of History and Theory of Architecture, Universitat Politècnica de Catalunya, 08034 Barcelona, Spain
*   Correspondence: zaidamuxim@gmail.com (Z.M.); arias.03@gmail.com (D.A.L.)

**Abstract:** After Francisco Franco's death, the process of democratisation of public institutions was a key factor in the evolution of the architectural profession in Spain. The approval of the creation of neighbourhood associations, the first municipal governments, and the modernisation of Spanish universities are some examples of this. Moreover, feminist and environmental activism from some parts of Spanish society was relevant for socio-political change that affected women in particular. The last decade of Franco's Regime coincided with the first generation of women that graduated from the Barcelona School of Architecture (ETSAB). From 1964 to 1975, 73 female students graduated as architects—the first one was Margarita Brender Rubira (1919–2000) who validated her degree obtained in Romania in 1962. Some of these women became pioneers in different fields of the architectural profession, such as Roser Amador in architectural design, Alrun Jimeno in building technologies, Anna Bofill in urban design and planning, Rosa Barba in landscape architecture or Pascuala Campos in architectural design, and teaching with gender perspective. This article presents the contributions of these women to the architecture profession in relation to these socio-political advances. It also seeks—through the life stories, personal experiences, and personal visions on professional practice—to highlight those 'other stories' that have been left out of the hegemonic historiography of Spanish architecture.

**Keywords:** female architects; pioneers; Barcelona; socio-political change; feminisms

## 1. Introduction Female Architects and Discrimination

Both historiographical narratives and oral accounts of architecture have discriminated against, excluded, omitted and rendered invisible female architects and designers. Discriminatory mechanisms in architectural practice and all the professions have varied origins, ranging from social constructs and features such as misogyny, the assignment of gender roles or the subordination of women, to those derived from the dominant systems (capitalism, Euro centrism, class, etc.,) or from scientific knowledge falsely constructed as neutral, objective and universal. Added to these still present mechanisms and prejudices, are the difficulties faced by women in reconciling their family and personal lives with their professional work. Those difficulties often cause women to view their family and professional lives as mutually exclusive, and to reject or postpone motherhood and child-rearing. A tactic that very few (if any) men would even consider.

According to the Valencian researcher and teacher of language and literature, Ana López-Navajas, the achievements of women are not included in the canonical accounts of history, nor those of the arts and sciences, and when they are accounted, it with a marked androcentric character and as an

exception (López-Navajas 2014). This has given rise to a profound historical bias, a single version of history that does not correspond to reality, resulting in a lack of role models and the perpetuation of stereotypes and false beliefs about professional women in general (Scott 1992). The lack of recognition and historical visibility generates the lack of female figures as referents. In architecture, a falsely imagined narrative is constructed in which female architects either did not exist, or if their existence was recognised, they were not considered good enough to stand out or be compared to the genius of the 'maestros' that have marked this history of architecture and the city.

This discrimination is not only evident in women's exclusion as historical subjects, but also in the function or eminence that historians have assigned them in the exercising of their profession. If they are described or named it is always as a subordinate figure whether it be as a draughtswoman, a secondary collaborator, inspirational muse, etc. This historiographical type-casting of the function, capacity and authorship, highlights the key components in the construction of the narrative: individuality and male heroism. Women represent the collective, as being part of the team. They are the generic, the indivisible, the invisible and their concealment also stems from the impertinence of having occupied areas of production that were not typical of or appropriate to the gender functions normally assigned.

The heroic account is strongly allied to the perpetuity of the discourses. When history consolidates and guarantees a single central male discourse it simultaneously produces and reproduces a refusal to include other contributions, whether these are new or previously included but not consolidated.

The scant representation of women architects in general terms relates to a lack of media visibility produced as a consequence of their otherness. It is not because they do not exist or because their production is not of sufficient quality to be published, rather that the determining factors for publication are the power relationships, the decision-making mechanisms, and the same consolidated and guaranteed standards that have been established by them. Thus, the personalities that have been most published and featured in the media, who are thereby most recognised and visible will be at the centre of discourses, disseminated as the models and over time consolidated into the sole experts or exemplar. Therefore, this article seeks to shine a light upon the first female architects who graduated from the Barcelona School of Architecture and their contributions in the era of transition to democracy.

## 2. Women, Feminism and the Socio-Political Context

> It has not just been our experience to deal with a casual and disguised misogyny, but the civil and criminal practices of the State and the whole of common morality ( . . . ). Ours is not a feminism of the lecture hall, but of lived experience. First came anger and courage. The lectures came after. (Valcárcel and de Quirós 2000)

The first female Spanish architects graduated during the years of General Franco's dictatorship imposed at the end of the Spanish Civil War, 1936–1939. It was an era of silence and impunity as regards the basic civil rights of the population in general and women in particular. It is enough to recollect that women at that time were not able to hold their own bank account or work without the express authorisation of the male head of the family. Feminist organisations began to make inroads in 1975[1] and formally establish themselves, although in the previous years a process of self-awareness had already commenced through groups from district and neighbourhood associations, in companies and in universities. As Nuria Varela explains, there was a sense of urgency to destroy the model of femininity that Franco's dictatorship had imposed (Varela 2013).

In 1964, during the last decade of the dictatorship (1939–1975), the Law of Associations was passed, which legalised the neighbourhood associations, providing, although hardly with the regime's

---

[1] In February 1975, the Platform of Feminist Organisations, which had naturally consolidated, organised a press conference in a pub in the centre of Madrid. At the international level, the United Nations had made two calls for this International Women's Year: the World Conference, -government-, which was held in Mexico from 19 June to 2 July and the World Women's Congress, aimed at non-governmental organisations, which took place in East Berlin from 20 to 24 October of the same year.

blessing, the first urban spaces for democratic discussion and thinking. These associations contained women's committees that strove to highlight the difficulties of daily life in the late Franco-era city, calling for urban improvements, services, facilities and public spaces for the betterment of everyday life. This call is a consequence of the division of jobs based on gender roles, as a result of which women are mainly left to take charge of reproductive tasks that give them a broad understanding of the necessities that a city must satisfy and from which traditional urban planning was far removed. In 1970, a study was carried out in the working class district of Besós in Barcelona to understand the needs for nurseries, and which discovered that 22% of women worked outside the home and 50% worked for a living within their own homes. Obviously, a feature of life at that time was that community and social services enabling women to escape from the obligations of child rearing and caring hardly existed.

Anything that was needed had to come from their own experience and circumstances, as it was not until 1975 when the first texts relating to European and North American feminism began to arrive. The elaboration of feminist theories in Spain as in the rest of the world was centred on a criticism of maternity, marriage, family and the sexual model of a politically endorsed patriarchal society. The central theme was the fight for the revocation of all discriminatory laws against women in all areas of their lives.[2]

Thus, feminist movements took on great relevance in the years of the Spanish transition by arguing for the incorporation of everyday life into the practices of urban planning. Feminist groups organised the 'First Women's Liberation Days' in Madrid in December 1975 and, in March 1976, 'Catalan Women's Days' were celebrated in Barcelona. The Barcelona event was attended by three thousand people with representation from all over Spain and participation from the neighbourhood movements, through the attendance of ten associations, and nineteen women's delegates from Catalan neighbourhood associations (Muxí and Magro 2009).

One of the nine reports presented, within the section talks, Communications and Conclusions, was 'Women and Neighbourhoods', in which the existing relations between women and their local environment was discussed and revealed, bearing in mind that this was the place where their everyday life happened. Alongside this report a series of talks were presented which dealt with the same topic adding more points of view and experiences.

The report 'Women and Neighbourhoods', [ . . . ] tackled two important questions . . . [ . . . ] The poor living conditions in the neighbourhoods affect women most of all as it is they who spend most time there performing most of their tasks [and] The problem of the people's participation is more serious in a woman's case because she finds herself in a position of inferiority, because of the oppression she has suffered and her exclusion from the public space, compared to the man (Muxí and Magro 2009).

The journal *Vindicación feminista* (1976–1979) published in its edition number 4, of October 1976, a new section 'The woman in the neighbourhoods' with the objective of revealing 'as faithfully as possible the conditions in which thousands of women lived in the neighbourhoods, the forms of oppression which women suffered, and the daily, brutal or subtly discriminatory realities of life'.

In 1980 the Independent Feminist Days were celebrated in Barcelona. As part of the event, Anna Bofill Levy gave a talk on 'Women and Architecture', a different female viewpoint on the urban environment revealing the relations between patriarchal structures and the shape of our cities.

Meanwhile, Pascuala Campos de Michelena, who had already come to believe that men made planning decisions in accordance with their own interests, had joined the initial group of independent Galician feminists (FIGA) at the end of the 1970s. Her preoccupation with the analysis of constructed space and reflection on feminism and architecture went hand in hand with her personal life experiences. She participated in the organisation of the Feminist Days in Vigo and attended others in San Sebastian and Granada. She also attended the International Congress of Female Architects in Paris and Berlin

---

2　In 1976, two struggles continued throughout 1977: amnesty and the decriminalization of adultery. The banners read: "Amnesty for crimes specific to the woman", "freedom of women prisoners", "contraception, abortion, prostitution, adultery are not crimes. Amnesty".

and took part in the Permanent Seminar Women and the City, in which she co-directed, the course on 'Town Planning and Women'. New visions of private and public spaces.

## 3. The First Female Architects from the Barcelona School of Architecture

In Spain, the first degrees in architecture were awarded in 1757 from the Academy of San Fernando in Madrid as part of courses in painting, sculpture and architecture. From 1847 Special Studies in Architecture were created that evolved until in 1857 when the School of Architecture was created, currently known as ETSAM. In Barcelona the *Clase de Arquitectura* existed between 1817 and 1850 whose students had to be ratified by the Academy of San Fernando. It operated in the School of Master Builders between 1850 and 1870 until 1875 when the Barcelona School of Architecture was created.

The Royal Decree of 8 March 1810 established the equality of women with men for entry into university (Sáenz Berceo 2010). However, women had to wait until 1936 to graduate as architects in Spain. The first woman to do so that same year from ETSAM (Madrid School of architecture) was Matilde Ucelay Maórtua (1912–2008), Rita Fernández Queimadelos followed (1911–2008) in 1940, and up until 1960 with Milagros Rey Hombre no other female architect graduated from the school. In 1962 the city of Barcelona awarded its first architecture degree to a woman. The recipient was Margarita Bender Rubira, but she only ratified studies that had already been realised. In the ETSAB Barcelona School of Architecture, the first female architect to graduate was Mercedes Serra Barenys, who obtained her degree in 1964.

As we previously mentioned, the period prior to the transition to democracy was denoted for its sexism and gynopia.[3] The situation was particularly dominated by an incapacity to fully recognise women and the disavowal of the feminine experience, which was camouflaged under the supposed equality right of access to university. Franco's dictatorship had destroyed the female references. The heirs of Concepción Arenal, Spanish feminist pioneer, who had trod the university's halls dressed as a man, began to fill the same halls ignorant of whose footsteps they were treading in.

By 1975 ETSAB was celebrating its centenary, and eleven years after awarding its first degree to a woman, 73 more women had obtained the qualification of '*arquitecto*'. But women had to wait until 2005 before they received the title of '*arquitecta*' (Criteris Lingüístics UPC n.d.).[4] In the same year the number of male and female students reached parity. In 2018 the number of female students continued around 50%, although for the 2017–2018 cohort the number of female graduates was slightly above male architects (96 vs. 87, and 122 vs. 111, for degrees and master's degrees respectively), however, neither professional recognition nor their presence in the academic ranks reflected the same proportions. In 2018, female academics constituted 31% of the teaching staff. A statistical analysis of current teaching staff reveals that in the most recent categories of '*Agregado*' and 'Lecturer' there is a growing incorporation of women. Women make up 43.5% of the ranks of 'Teaching Assistant' and 45.5% of the numbers of Lecturers. Only two professors are women compared with 18 men. Despite the inequality that still exists it is important to point out that in 2005 female teachers formed just 17% of the staff, whereas 13 years later the numbers of female teachers had increased to 31%.

For those *arquitectas* that graduated between 1964 and 1975, academic life was marked by being the first women to study architecture, often being the only woman in classes of men, the high level of

---

[3] Ginopia is a neologism used in legal matters as a way of naming the omission of the woman's point of view in gender violence cases. Also, in the Feminist movement this term is used as a definition of symbolic violence, where omission is used as a manifestation of abuse. The ginopia is closely related to the androcentric culture, to power, to the struggle for the preservation of undeserved privileges, to the culture of domination, among many other aspects. The definition was provided by Marina Morelli Núñez, Ph.D. in Law and Social Sciences, Uruguay. Available at: http://www.larepublica.com.uy/mujeres/319436-ginopia.

[4] In the Spanish language, the word "arquitecto" is used to refer to the male architect and that is how the diplomas issued by the University of that time were written. It would not be until 2005, with the approval and application of the linguistic criteria of gender by the Polytechnic University of Catalonia, that women architects would be designated as "arquitectas" (Agreement 272/2005 of the Governing Council. Criteria updated and ratified by the Equal Opportunities Commission in 2011).

competition, studying with highly recognised architects and a political awakening to society in general and the school in particular. The politicisation of the school was both a response to the social-political changes occurring abroad, such as the events of May 1968 in France, and also internal politics that generated a wave of strikes, the closure of the school and the expulsion of the teachers. Many of the women interviewed in 'Arquitectura en femenino' (Muxí and Covaleda 2013) think that they have passed through the school having learned little, although with a heavy dosage of political and social commitment.

In this period the ETSAB curriculum was changed three times. The final change came in 1973, therefore the group of women analysed were affected by two plans in 1957 and 1964, both with a specialisation in town planning and buildings. Coinciding with the 1957 study plan, part of the teaching staff was renewed and classical academic teaching survived until 1956–1958, '[...] replaced by a faith, sometimes too hasty and immature, in so called modern architecture' (Domench i Girbau 1968, p. 29).

In the beginning of the 1960s the university offered certain social and democratic openings, which however did not last long. The preparation of the new plan of studies in 1964, which was conducted by a commission that mixed representatives of the Architects' Association of Catalonia and the Balearic Islands and the School of Architecture, incorporated social subjects such as economics and sociology. As these attempts to restructure teaching were being made the school opened its doors to international events such as the VIII International Conference of students of architecture that was held in summer 1963. International figures such as Ludovico Quaroni, Hugh Casson, Giancarlo de Carlo, and Jaap Bakema were invited to visit the school and wrote reports on its status (Domench i Girbau 1968). On 9 March 1966 a student meeting was held at the Capuchin Convent in Sarriá, Barcelona, with the aim of founding the Democratic Student Union of the University of Barcelona. There were 450 people gathered inside, when the police arrived to evict them, they refused because their identification was required, and they were welcomed by the monks. But on day 11, despite the mediation attempts of the religious community, the dictatorship ordered their removal and the identification of the men, although the 100 women present were not identified. As that was not a place for women to be, they were not recognised as 'revolutionary' or politically involved, because that was an action not for women. So, they were saved from being recognised but it was a way to not recognise their political rights to participate. The teachers and students who participated were later expelled. The incident would mark the end of the brief experience of opening at the school, as well as at the architects' association, as the association bodies turned against the progressive wing, and the usual atmosphere of repression would return and endure until the end of the dictatorship.

## 4. The Pioneers: Diversity of Professional Practice

In the social and political context described above, came other accounts and female expressions of architecture, voices emanating from the Barcelona School of Architecture (ETSAB), lost or omitted under the influence of the dominant accounts of the postmodern period and in contrast to the prominence granted to male architects. Because of the hidden and secondary position of these other experiences it is only possible to identify them through specific research, recent biographies or through connections with partners who have been featured in the various publications.

These contributions to architecture and the city have been characterised by a variety of perspectives, practices and areas of study, in addition to different forms of approach. An illustration of this is Alrun Jimeno Urban (Vienna, 1941), the third female architect to graduate from ETSAB in 1966, immediately after her graduation she began to teach courses in interior design in the Elisava and Llar Schools. Adapting herself to her husband's professional career, she spent five years in Seville where she taught courses in technical drawing, combining her teaching with studio work. One of her first projects of her own was an extension to the German School (Colegio Alemán) in Seville. On her return to Barcelona she worked in an academic capacity at ETSAB, once again juggling her teaching with various projects that she undertook personally: a tennis club and a swimming pool, a housing block, a country house

and project management for the Cruzcampo brewery. In 1985, she formed part of the governing board of the Architects' Association of Catalonia (COAC), for whom she organised, among other tasks, an exhibition and catalogue to celebrate one hundred years of the Catalan Association in Buenos Aires. She obtained her doctorate in 1993 with the thesis 'Approaching the work of Francesc Mitjans'. She was a tenured lecturer in the Department of Building at the University School of ETSAB until her retirement in 2006.

Like Alrun Jimeno Urban, the architect Pascuala Campos de Michelena (Sabiote, Jaén, 1938), who also graduated from ETSAB in 1966, centred her career on teaching (Quixal 2015). She was a lecturer from 1980 onwards in the School of Architecture of A Coruña, created in 1975, becoming the first chair of Architectural Projects in all of Spain's schools with a work entitled Space and Gender (*Espazo e Xénero).* In addition to being an architect, lecturer, and mother, Pascuala Campos has been an active feminist and pioneer in the inclusion of gender perspectives in the teaching of architecture, directing and participating in courses and seminars on the theme. She co-directed courses on 'New visions of public and private space', alongside Adriana Bisquet, María Ángeles Durán and Rosa Barba, as well as courses on 'Body, space and architecture'.

In her first years as an architect, she worked as a planner like Alrum Jimeno. She moved to Pontevedra, Galicia, with her husband, the architect César Portela, establishing an associate studio of architecture. Works that stand out from this period include a housing complex for gypsies in Campañó, Poio (1971–1973) and Bueu market (1971–1972). Both were selected for the travelling exhibition 'Architecture and Rationalism. Aldo Rossi + 21 Spanish Architects' between 1975 and October 1976. Other works of this period include an apartment block in the Campolongo estate, Pontevedra (1973) and the Pontecesures council buildings, A Coruña (1973–1975) and Forcarei, Pontevedra (1974–1980).

In addition to teaching, another field of activity, although undervalued and little publicised, is the presence of women on editorial boards and as frequent editors of articles in architectural periodicals and journals. This is illustrated by the career of Roser Amador Cercós (Barcelona, 1944), who graduated in 1968 as the sixth female architect, and was editor of the journal *Nuevo Ambiente* in her first years as a practising professional. Under her direction, throughout 1972, and the first issue of 1973 produced together with Marta Ribalta, the journal contained some important transformations: changes to the front cover, recognition of the authors of the projects presented, incorporation of a summary of numbered contents, articles with less descriptive contents and longer pieces by and about local architects. From 1974 onwards she worked with Lluís Domènech Girbau, in 1980 they founded the studio Amadó-Domènech arquitectes which was reconfigured with a change of name in 2001, and would become the company B01 Arquitectes, S.L.P. This change of name indicates the lack of individual ego that characterised the team. We could say that it is a feminine characteristic to recognise the work of the group.

In 1977 the exhibition 'Architecture for after a war' organised by the Architects' Association of Catalonia and the Balearic Islands had a catalogue with three texts, one by Roser Amadó and Lluis Domenech together with one by Antón Capitel and Carlos Sambricio.

Roser Amadó's postgraduate training had included a specialisation in heritage restoration in the 1980s and Master MAUS (Spanish acronym for urban environment and sustainability) taken in 1998 at the UPC (Polytechnic University of Catalonia). This approach to urban planning and heritage themes distinguished the years after the Amadó-Domènech company. In 1980 they carried out the Plan for the Historic Centre of Lleida, with the collaboration of the architects Joan Busquets and Ramón Puig. The main aim of the project was to find a planning solution for the deterioration of the city centre, made more complex by its topographic characteristics. In 1985 they obtained the National Prize for Urban Planning and were also published in numerous books and journals both in Spain and internationally (Marciani 2015).

Their restorations were distinguished by a respect for the pre-existing conditions of the site that were transformed in a contemporary way and by adding seamless layers in the design form.

A fine example of this is the building Casa de L'Ardiaca (1990–1997) in which they used light, as they had in other restorations, as one of the principal elements to aerate the building and make it breathe.

Subsequently Roser Amadó and Lluis Domènech studied the architecture of Domènech i Montaner. This allowed them to intervene in various works of this modernist architect, such as the former publishing house Montaner i Simón (1879) that they rehabilitated in 1991 as the current Fundación Tàpies. In this work they sought to return the building to its original condition, by exploiting its richness of space, respecting the facade and adapting the interior in order to create a museum. For this work Amadó - Domènech received the first Década prize (2001), awarded by a jury made up of Robert Venturi and Denise Scott-Brown.

Other areas of architectural practice, not usually traditionally considered and honoured as relevant by the hegemonic history, is illustrated by the achievements of Rosa Barba (Barcelona, 1948–2000) who developed urban and country planning and landscaping projects, becoming an expert and key figure both in the academic and professional environments.

Rosa Barba obtained a degree in architecture in 1971 and like the previously mentioned female architects began to work immediately after graduation. She founded the studio *Rosa Barba & Ricard Pié arquitectes* in 1971. Her professional work began in 1967 collaborating in the Highways Plan and the Barcelona District Plan. After her graduation, she produced a theoretical proposal for the new city of Riera de Calders. She collaborated in studies and projects for various neighbourhoods and communities in Catalonia. She carried out a development study on the Catalonian coastline and the esplanade from Sant Antoni de Calonge and Gran Canaria and Tenerife Island Plan. In Mallorca she produced the General Urban Development Plan for Soller and the Special Plan for the Internal Reform of the historic centre of Pollença. In the Costa Brava she carried out studies and works for the General Urban Development Plans of Torroella de Montgrí and Parc dels Estanys en Platja d'Aro. In 1998 she carried out a project for 22 dwellings in Horta-Guinardó, Barcelona, and in 1999 completed a project for 60 dwellings in Vallbona, Barcelona ([Rosero 2015](#)).

Her pioneering career as a teacher and researcher in the area of landscape is remarkable, creating a school of interpretation and design of the Mediterranean landscape. In 1987 she obtained a doctorate with the thesis *The abstraction of the territory*, mentored by Manuel de Solà-Morales, being awarded a cum laude and receiving an extraordinary prize from the Polytechnic University of Catalonia (UPC). Two years later, in 1989 she became a tenured lecturer, having begun her lecturing career in 1974. In 1992 she was appointed director of the Master's Degree in Landscaping of the UPC, which had been founded and directed by Manuel Ribas Piera in 1983, and which obtained the certification from the *European Federation for Landscape Architecture*. In addition to the ETSAB, she gave classes and talks in various cities such as Cambridge, Vienna, Versailles, Venice, Rome and Madrid. In 1993 she became a founder member of the Centre for Research into Landscape Projects, a university-based research project focused on the transfer of knowledge and training from experts in the area of landscape. In the same year the ETSAB higher Degree in Landscaping was established.

Rosa Barba was also prolific in her published works. The book *Rosa Barba Casanovas, 1970–2000: Works and Writings* contains a collection of her works and articles, at the time that her colleagues at ETSAB Ricard Pié, Josep M Vilanova, Maria Goula and Jordi Bellmunt explain several of their projects and ideas. The book highlights three of the research projects that featured in her career: the regulation of colour in the restoration of quarries; a study on the Madrid city park Manzanares Sur and the European project Artemis. As regards her publications, these include: *Arguments in the design of landscape*; *The design of the place*; *Projection, in what landscape*?; *Barcelona Open Spaces Plan* and *Papers on Landscape*. Further articles can be found in journals such as *Quaderns d' Arquitectura*; *Estudis urbans*; *L'art de ben establir* ([Solà-Morales 1983](#)); *Geometría*; and others ([Barba 1992](#), [1996](#)). Among her books as an editor can be found *Architecture and Tourism: Plans and Projects* (1996), *Restoration and Interventions, Landscape in the quarries of Minorca* (1998).

　　　Rosa Barba was one of the main promoters of the European Biennial of Landscape. In recognition of her illustrious career and for being a Spanish exemplar in the discipline, in 2001 the Architects' Association of Catalonia named the biannual European Landscape Prize, instituted in 1999, in her honour.

　　　In a similar way to how Rosa Barba stood out in seeking to promote and defend landscape studies in Catalonia, as a means of understanding physical reality and its transformation (distinct from the Anglo-Saxon tradition), Anna Bofill Levi (Barcelona, 1944) is an essential pioneer of feminism in architecture, who has promoted and reflected widely on town planning based on women's experiences.

　　　Anna Bofill was the 28th woman to graduate from ETSAB, completing her studies in 1972, right at the peak of feminism's third wave. She obtained her doctorate in 1975 with the thesis *Contribución al estudio de la generación geométrica de formas arquitectónicas y urbanas* (Contribution to the study of the geometric generation of architectural and urban forms).

　　　Her most important architectural works include the apartment block Walden 7 carried out when she formed part of the *Taller de Arquitectura* and was sole author of the Plaza Cataluña railway station in Barcelona. Her career as an architect can be divided into two different stages.

　　　The first stage, from the mid-1960s until 1982, was her work as part of the *Taller de Arquitectura* team that constituted a unique experience working with a multidisciplinary team of musicians, poets, architects and photographers who wanted to change the world.[5] The *Taller de Arquitectura* carried out research into architecture and town planning, making alternative proposals for social housing by considering other ways of living. The *Taller* carried out numerous theoretical projects and also constructed the Barrio Gaudí in Reus and a building of more than 400 housing units called Walden 7 in Sant Just Desvern. This last building, in which the *arquitecta* herself lives, retained an unsurpassed look of modernity, especially in the design of the dwellings, the geometric articulation of the complex, the flexibility and the communal spaces. The Walden 7 project 'aimed to break', in the words of Anna Bofill, with the 'antisocial and functionalist residential block advocated by the CIAM (Congrès Internationaux d'Architecture Moderne)' (Gómez Moriana 2018). To achieve this objective, the *Taller* was inspired by the vernacular Mediterranean coasts towns, extrapolating their intricate urban structure to a three-dimensional level through mathematical algorithms developed by Anna Bofill and explained in her 1974 Ph.D.

　　　However, the historiography of architecture has made her and the team's contributions invisible, suggesting that her brother Ricardo Bofill is the main author. Despite being a team of many people, it is difficult to find the names of the women who were involved since the accounts of this experience have always sought to identify the hero, the captain, not understanding that in the beginning it was a truly collective experience. Anna Bofill's personal life experience gradually made her see that this group had a glass ceiling and that her transformative designs were not acceptable in a studio that had already become international, and less experimental, with an emphasis on individual authorship. Therefore, she abandoned the studio in 1982 (divorcing at the same time) and so began her second stage as a female architect, committed to a vision based on feminism and the female experience of the city and architecture. She began working on incorporating the gender perspective into town planning.

　　　Working alone was not easy, as she explained in an interview that 'the position of a woman alone in a studio is very difficult if you do not set out from a financially favourable position which allows you to participate in competitions' (Muxí and Covaleda 2013). This position was even more difficult when you have children to look after, in her case two daughters. This second professional stage can be divided into two parts, one more classical involving architecture, building housing, facilities and the Plaza Cataluña railway station and the second, from the 1980s onwards, which she considered her greatest contribution, and gave her most cause for pride, which was her research and designs for town

---

5　　The initial team was formed by Ricardo Bofill, Anna Bofill, Salvador Clotas, Ramón Collado, José Agustín Goytisolo, Joan Malagarriga, Manuel Núñez Yanowsky, Dolors Rocamora, and Serena Vergano.

planning based on women's experiences. In this field of town planning one of her most influential publications was *Libro blanco. Las mujeres y la ciudad* (White paper. Women and the City) co-authored with Rosa María Dumenjó Martí and Isabel Segura Soriano, for the European Union through the Fundació Maria Aurelia Capmany in 1998.

Anna Bofill is also a composer, a member of the Association of Women in Music and the Catalan Association of Composers, in addition to participating in various projects around women in the History of Music.

## 5. Final Reflections

All the achievements of the pioneering female architects mentioned above, who had played leading professional roles at a time of major political and social change, were honoured with prizes after long careers. Their contributions, whether theoretical or constructed works, (some produced independently, others with partners) were, as has been shown, published in periodical journals or books of articles on Spanish architecture. Yet their names and achievements, in contrast to their male colleagues, have not been given due and equal recognition, either in the historical record, or in memory.

As we reflect upon them, we may observe that in spite of the remarkable work of these female architects, graduates of the Barcelona School of Architecture, none of them has attained the prominence that they deserve from the academy, the publishing houses or their own professional bodies.

Despite the ever-increasing and invaluable work performed by a number of teams (groups with institutional, academic or educational objectives organised mainly by female architects) with the aim of bringing into memory the work of so many professional women, the names of the *arquitectas* recounted here still do not form part of the curriculum of architectural schools at state level. Perhaps it is because the adjustment processes of the academy and those who comprise it continue to lag behind social changes in general. Or perhaps because the single hegemonic history rooted in the values of heroic masculine individuality does not allow such inclusion, not only of the presence of women, but other models of action and practice, distinct, alternative and diverse. The lack of prominence and exclusion of women are not only for being women but for not following the established patterns in ways of working or of the chosen fields.

Of all the female architects mentioned here, Roser Amadó is the first woman to be included in a history book of Spanish architecture: *Spanish Architecture 1950 to 1980* written by Antón Capitel. Roser Amadó and Lluís Domènech have always collaborated and operated as a team with shared authorship and both their signatures can be seen in their projects and various articles (González Capitel 1986). This inclusion reflects Anton Capitel's acquaintance and close collaboration with both of them in the previously cited exhibition 'Architecture for after a war'. This fact implies a difference to internationally recognised historians such as Sigfried Giedion for example, who was personally acquainted with the companies of Alvar Aalto and his first partner Aino Marsio and later with Elsa Kaisa Mäkiniemi, but who, nevertheless deliberately excluded them from the historical account.[6] In 1933 the Aaltos sent the drawings and models of the Paimio sanatorium to Athens for the exhibition to be held during the CIAM. As an acknowledgement of their receipt Giedion sent a postcard[7] to the architect in which he wrote: 'Your sanatorium has arrived. One can sense your (or Aino's?) hand in every plate!' (Pelkonen 2014). This note suggests that, although Aino Marsio's authorship was excluded from his textbooks, the professional collective of the time in general, and Giedion in particular, recognised teamwork to the point of not distinguishing authorships individually.

---

[6]　In the first edition of *Space, Time and Architecture* of 1941 Aino Marsio is omitted from the narration. After her death in 1949, the historian Sigfried Giedion added her to one of his later revisions. It is significant, as it points out how fundamental Aino Marsio was in the life and work of Aalto as a wife and partner but does not make any reference to her work (nor Artek) in the corpus of the text, either in notes, or in images.

[7]　Postcard sent by Giedion to Alvar Aalto, written in Zurich, 6 August 1933.

We also observe that these female architects threw themselves into the world of work immediately after graduation, and despite the assigned gender roles, the architects described here have all been mothers, and the majority of them continued studying in order to acquire specialisations and academic degrees. The empowerment of women in general in the 1970s and the opening created by the emergence of democracy in the 1980s provided a context which enabled and promoted this personal and professional development.

To this day these absences pose questions to us as architects and as a society. We must question and challenge the conceptual framework for the organisation of traditional knowledge, because they have left out the experiences, activities and ideas of more than half of humanity.

**Author Contributions:** Z.M.'s main contribution to this article is the seminal research about the first women architect graduated in Barcelona School of Architecture. D.A.L.'s main contribution comes from her Ph.D. research (http://hdl.handle.net/2117/123109) about the process to render invisible the women architect's work in the modern movement's historiography. All authors have read and agreed to the published version of the manuscript.

**Funding:** This research was funded by the Ministry of Science, Innovation, and Universities, Spanish Government. Research Project Title: Women in Spanish (Post)Modern Architecture Culture, 1965–2000. Grant number: PGC2018-095905-A-I00.

**Conflicts of Interest:** The authors declare no conflict of interest.

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
