# Peer review of "Filling History, Consolidating the Origins. The First Female Architects of the Barcelona School of Architecture (1964–1975)"

_arts, 1939_

Round 1

Reviewer 1 Report

This is an interesting paper delving into the careers of some of the female graduates of the Barcelona School of Architecture at a particularly turbulent time in the history of Spain and Catalunya. It adds to the story of the mechanisms by which women have been excluded from the canon of architecture despite exemplary work. The paper seeks to "shine a light" on this generation of pioneer women architects and demonstrate how their careers have not been fully acknowledged or accredited. Their contributions to the transition to democracy are less well explored/explained, but that transition does provide an interesting background.

The introduction might be better framed as discussing the global problem of the exclusion of women from accepted histories (because it is global) and then demonstrate precisely how that happens in the careers of the graduates.  

English expression and grammar is sometimes clumsy and needs work. Sentences are at times too long and become convoluted and difficult to understand. For example: the sentence from lines 29-33 might be better as three sentences ending with "A tactic that very few (if any) men would even consider."

The use of the word 'referent' is very unclear - it has a particular meaning in English that does not seem appropriate in the text. Perhaps role model might be a better choice, at others points expert or pioneer or exemplar, key figure might be better. The paper needs a thorough edit for language.

Also, quotation marks are a bit random in the paragraphs contained in 108-117. Are these two paragraphs full quotations? Should they be indented? Line 119 has a " without the closing ".

Please check dates for Matilde Ucelay Maórtua in line 143 - I presume it should be 1912 not 1812.

Line 169: the numbers of female teachers did not increase by 14%. You could say the proportion increased by 14 percentage points but probably better to repeat the figure and say: increased to 31%. Or could say the proportion nearly doubled to 31%.

Line 176 - the reference is not listed in the list of references at the end of the paper.

Paragraph beginning line 185 has some switches in tense which need fixing. I also wonder if the story of the meeting in the Capuchin Convent is needed. Or if included, it would be good to know whether there were many architecture students and teachers in the group. I was also curious as to whether the women who were present were expelled, given they were not identified. Was this a rare event when being female actually helped?

Line 250, the words "individual ego" would be better than personalism.

The section on Anna Bofill might be interesting to further unpick the tensions of attribution or authorship of architecture that the career of Roser Armado highlights. Why is it that her brother Ricardo's name is globally known and hers not? Taller de Arquitectura began as a multi-disciplinary team but... While I can understand why it might be a positive act to talk of her independent of him, I think the tension might be also fruitful. Anna's work seems like it was critical to the success of the early Taller work like Walden 7, is her PhD not about the process that developed Walden 7? And Taller's subsequent work seems to disavow the radicalism of the project, getting more and more grandiose over the years. You do speak of that, but I think it could be teased out more. Is it that those who write architectural history need a name to hang buildings on, need a sole (male) author? Others have written about this and it would be good to tie in this example with that wider material.

Author Response

The authors consider your review as relevant and we have modified the manuscript according to your comments and suggestions

Reviewer 2 Report

This study on the Spanish (more specific: Catalan) situation of the role of women in postwar architecture education is a needed piece in the larger picture of rewriting and rethinking architectural history, the profession and the models of practice. Understandably each region and country faced different challenges, yet the end of the Franco dictatorship and transition period into parliamentary democracy seem to make the case historically relevant.

The author(s) contextualise the Spanisch/Catalan situation and describe the phase 1964–75 which saw the first female graduates from architecture schools. And the focus on a few selected biographies (or trajectories) of these female graduates seems successful. However, there is a number of problems with sources (translated titles of articles, books, exhibitions without the Spanish original footnote or Spanish titles and explanations without English translations, etc. – see file attached). Also, on a methodological level, it remains unclear, how the Spanish/Catalan example compares to other countries (and my suggestion would be to refer to other national studies on female graduates in other European countries, such as France, Italy, Germany or the Scandinavian countries, or for that matter, maybe compare the situation in Latin America? To my knowledge, there are female architects in Cuba before 1964)

Otherwise the significance of this Spanish/Catalan example remains unclear, and the reference to Giedion's practice of writing women out of the work produced by architects couples does not make up for this question of relevance, but rather open a new topic altogether. 

Author Response

The authors consider your review as relevant and we have modified the manuscript according to your comments and suggestions.

Round 2

Reviewer 1 Report

The paper stills requires a thorough edit for English language use and expression, but I think that would be a job for the editors - the authors having done some work on this.